# Can the Standard Configuration of a Cardiac Monitor Lead to Medical Errors under a Stress Induction?

**DOI:** 10.3390/s22093536

**Published:** 2022-05-06

**Authors:** Maja Dzisko, Anna Lewandowska, Beata Wudarska

**Affiliations:** 1Computer Science and Information Technology, West Pomeranian University of Technology, 70-310 Szczecin, Poland; atomaszewska@wi.zut.edu.pl; 2Western Pomerania District Hospital, Zdunowo, 70-890 Szczecin, Poland; bwudarska@eurescue.com

**Keywords:** vital sign monitor, medical interface, eye-tracking metrics, stress, computer vision, ECG signal

## Abstract

The essential factor that enables medical patient monitoring is the vital signs monitor, whereas the key in communication with the monitor is the user interface. The way the data display on the monitors is standard, and it is often not changed; however, vital signs monitors are now configurable. Not all the data have to be displayed concurrently; not all data are necessary at a given moment. There arises a question: is the standard monitor configuration sufficient, or can it lead to mistakes related to delays in perceiving parameter changes? Some researchers argue that mistakes in life-saving activities is not mainly due to medical mistakes but due to poorly designed patient life monitor interfaces, among other reasons. In addition, it should be emphasized that the activity that saves the patient’s life is accompanied by stress, which is often caused by the chaos occurring in the hospital emergency department. This raises the following question: is the standard user interface, which they are used to, still effective under stress conditions? Therefore, our primary consideration is the measure of reaction speed of medical staff, which means the perception of the changes of vital signs on the patient’s monitor, for stress and stressless situations. The paper attempts to test the thesis of the importance of the medical interface and its relation to medical mistakes, extending it with knowledge about the difference in speed of making decisions by the medical staff with regard to the stress stimulus.

## 1. Introduction

The graphic user interface is now an inseparable element of the work of almost every human being. Paying particular attention to the characteristics of people’s work in the medical industry, it is noticeable that their work, especially of doctors and nurses, often occurs in the general chaos caused by a wide variety of emergencies connected with their patients. This increases the level of fatigue and stress in physicians.

One of the essential tools used by medical staff is patient life monitors, which started to be common in the 1980s [1]. Electronic monitoring facilitated the human senses measurements (inspired oxygen measurement, capnography, and pulse oximetry) and allowed genuine real-time continuous monitoring of oxygen delivery and patient ventilation and oxygenation. However, cardiac monitors were pushed down only to the initial level of patient examination—triage despite such facility. Moreover, the marriage between that technological product and the human user is not flawless. Still, the intensive care unit (ICU) or emergency room (ER) environments lack an individualized approach: user-tailored interfaces. Even hospital IT systems are not designed to reflect the cognitive processes underpinning clinical work [1]. They can influence the situation where the type of display of clinical data may cause a shift from correct to incorrect decision [2], or at least to a correct but delayed decision, taken by medical staff.

The monitors noted above are graphical interfaces designed to display vital signs. Unfortunately, to this day, patient monitors are not perfect, so not every change in patient parameters is immediately noticed [3]. According to Fairbanks and Caplan in [4], many medical adverse events are the result of poor interface design, not human error. It turns out that incorrectly displayed messages on the patient’s life monitor mean that the doctor does not always notice them. It often turns out that the displayed message is not of the right size, so the medical staff will not be able to catch it from a distance [3]. As noted above, stress or mental fatigue are crucial factors influencing this.

Therefore, it is crucial that medical staff work must be supported by the interface, not hindered. It is worth noting that, currently, vital signs patient’s monitors are configurable [5]. Not all the data have to be displayed concurrently, and not all data are necessary at a given moment. There arises a question: does the standard monitor configuration that is the most often used sufficient, or can it lead to mistakes related to delays in perceiving vital signs changes? Therefore, it is worth analyzing the effectiveness of patient life monitors with a standard user interface setting, especially under a stressful situation. Previous research, which pointed to the relationship of medical errors with the user interface, mainly concerned the lack of correct feedback for the user interface [4,6,7]. However, these studies did not discuss the influence of a stress factor on the efficiency of interaction between medical staff and the cardiac monitor interface.

As stress and fatigue are integral components of the work of medics, the medical graphical user interfaces useful in a neutral (stressless) situation may not be compatible in rapid and emergency situations when users’ reactions can be fairly different. The problem is presented in Figure 1, where we indicate the possibility of differences in the reaction time of medical personnel to the decrease in the patient’s vital parameters in a neutral (stress-free) and stressful situation. That is, there may be different perception and observation of changes in the patient’s vital parameters depending on the occurrence of the stressors. If such a situation occurs, the cardio-monitor’s standard-setting is ineffective, and the physicians’ perception depends on the conditions prevailing during the monitoring of the patient’s condition.

This article attempts to examine medical staff’s reactions in a stressful situation when working with the patient’s life monitor and answer the following question: is the standard monitor configuration sufficient, or can it lead to mistakes related to delays in perceiving parameters changes? Therefore we formulate the following *Hypothesis*: The visual detection of critical parameters under stress is less noticeable than during stress-free situations. Thus, the main goal of our research was to check the reaction of medical staff in a stressful situation related to the perception of the patient’s vital parameters. To achieve our aim, we designed the user study with medical simulators to verify the given hypothesis precisely with a virtual patient simulator and its life monitor.

Our research is based primarily on the analysis of the medics’ eyes movements, i.e., which places of the patient’s life monitor are under the interest of medics in a stressful and stress-free situation. In order to be sure that the stimuli that the participants were subjected to indeed caused stress, we analyzed not only the subject’s eye movement with eye-tracker (ET signal) but also the heart rate record with a wearable chest belt for heart monitoring (ECG signal). As both ECG signal metrics (such as HR (ECG) and LF/HF (ECG)) and ET (such as pupil size and time of fixation) identify a stressful situation, they have been used in our analysis.

The main contributions of our work are:Measuring medical staff reactions related to perception of vital signs on patient’s monitor under stress and stress-free situations;Confirming the efficiency of the stressors used in the experiment (that they really induce stress) through the metrics computed from acquired ECG and ET signals;Drawing attention to the problem related to the current functional layout of the patient’s graphical interfaces.

In order to achieve our research goal, an experiment was conducted. We describe it in Section 3. The analysis of the received data (ECG and ET signals) can be found in Section 4. Results are discussed in Section 5. Finally, we conclude the paper in Section 6.

## 2. Related Work

The effectiveness and reliability of an employee are primarily influenced by his/her adaptation to the working conditions. However, this adaptation is mainly possible without negative factors that make work difficult [8]. The problem is fundamental in medicine, as evidenced by the statistics that show inevitable medical errors [9]. Statistics prove that mortality rates are no longer the indicators of patient safety, as good hospitals may treat more sick or more severely traumatized patients and may present with worse survival rates despite efforts. Adverse events still occur in 3–4% of hospital stays, and of these, 25–50% are estimated to be avoidable. Obstacles include stress, unclear information, complex systems, interruptions, multitasking, a lack of professional experience, and finally, poorly or at least not ideally designed medical appliances [9].

An important aspect is also that, since the early 1980s, numerous agencies and societies have addressed patient safety in medicine, e.g., Anaesthesia Patient Safety Foundation, Emergency Medicine Patient Safety Foundation, or National Patient Safety Agency [10]. Anaesthesia, intensive care, and emergency medicine constituted primary targets for improving patient safety due to life-threatening conditions of patients in these specialties, the need for prompt clinical decisions in stressful environments, and several mishaps, malpractices, and ‘near-miss events’ [10]. Moreover, according to Fairbanks et al. in [4], many medical errors are made by inadequate user interface design, which has a crucial impact on patient safety as it relates to giving a diagnosis.

Enrico Coiera, in [1], claimed that well-designed user interfaces ensure that decisions based on reading data from the interface should be safe. In contrast, incorrectly designed interfaces may, unfortunately, hurt the decision-making process, thus causing mistakes by the person using the interface. This approach could end up being detrimental to patients. Unfortunately, current generation clinical medical interfaces are designed with the default assumption that their users perform a single task and that their attention is entirely devoted to interacting with that interface. However, it should be noted here that the medical staff using the interface, which is the patient’s life monitor during life-saving activities, must focus both on what is displayed by the interface and on the patient at the same time. Thus, staff cannot devote their full attention to the medical interface alone. Therefore, the analysis of the efficiency of standard monitor settings and their influence on correct usage during stressful situations seems crucial.

The issue of problems with medical interfaces is addressed more widely in the literature. In [11], the authors refer to the interfaces in the context of medical alarms. They argue that more than six alerts in hospital wards can lead to an emergency risk that includes the misapplication of alarms. This can cause alarm fatigue and indifference to alarm reception. The authors also discuss alarm fatigue in [12]. Emergency fatigue, they argue, is a significant clinical problem, and the resulting fatigue may slow the perception of vital sign change. The fatigue issue with medical alarms was also raised by Cvach in [13]. The authors emphasize that severe injuries and death in patients often occur due to missed alarm events resulting from exhaustion of the medical staff with alarms. However, they did not analyze the user interface efficiency and the delay of response to the critical situations (that are signalled on the patient’s monitor) in an environment close to natural, especially under the stress condition.

Most of the publications on the problem of improper medical interface design mainly focused on propositions of solutions based on changes in the information architecture. According to a report [14], all physicians need to have standardized resources to understand better, classify, communicate, and prevent/avoid medical device misuse. Therefore, the authors decided to introduce a minimalist system that reduces the chaos in the user interface, thus allowing a faster reaction of the physician in the event of a threat to the patient’s life. Whereas, in report [15], the authors suggested a lack of appropriate tools for the design of fault-tolerant and ergonomic human-machine interfaces for medical devices. They designed their own system to analyze medics’ errors to improve the interfaces.

It is worth noting that mental overload makes people begin to behave less rationally (more chaotically) [16], which can lead to missing a critical alert on a patient’s life monitor. According to Salman et al. [17], a proper medical information system should enable reliable and accurate interaction between users and the system in an emergency. Therefore, in this paper, instead of improving the medical interface of the vital sign monitor of a patient, we focused on identifying the problem and analyzing the user interface efficiency in an environment close to the natural. Such an approach enables us to answer the following question: is the standard user interface that they are used to still effective under stress conditions? If not, the experiment’s procedure described in the paper should be repeated every time new settings of the patient’s monitor are proposed. It is crucial, as the settings of a cardio monitor can be efficient in a laboratory environment, however problematic, under stress conditions.

## 3. Materials and Method

The experiment aimed to test the medical staff’s time reaction time to the falling vital signs of the patient displayed on the patient’s life monitor, both in a stressful and stress-free situation. The experiment was carried out with 21 respondents who declared normal or corrected to normal vision. All analyzed data were fully anonymized. Before the experiment, the participants gave their informed written consent to use the data from the perceptual experiment in the research (according to the Bioethics Committee Agreement no KB-0012/24/2020).

As the experiment was performed using the medical vital sign monitor, eye tracker, and wearable ECG, the task of the respondents was to connect the patient to medical equipment and then to monitor the patient’s vital signs. Before starting the experiment, the respondent was informed that in a case of falling the vital signs of the patient she/he was obliged to undertake an appropriate medical procedure aimed at saving the patient’s life. The person responsible for the control of the proper experiment conduct had the possibility to change the patients’ vital parameters, i.e., decreasing them in a given time.

**Test stand.** For the experiment we used the test stand located in the Medical Simulation Center of Collegium Medicum of the University of Zielona Gora (see Figure 2). The test stand consisted of a patient’s dummy, necessary medical devices to monitor the patient’s life, such as a patient’s life monitor—Terra Monitor, an ECG, and a pulse oximeter. In addition, we used the Tobii Eye Tracker Glasses 2 [18], which enabled recording the perception of the environment by the user during the experimental procedure. To ensure that the stressors we used in the experiment induced stress properly, the users wore an AidLab ECG chest belt [19] that recorded ECG signal (see Figure 2). It is worth noting that the laboratory conditions closely resembled natural states, which allowed us to reproduce a hospital room.

**Choice of stimulus.** Two types of stressors were applied to induce stress in the subject during the experiment. We used the sound of a ringing telephone and the signal of an ambulance. Choice of the stressors was motivated by simulation of natural conditions occurring in a hospital emergency department. The stressors were applied randomly.

**Experimental procedure**. The experiment was divided into two parts: Experiment run 1 and Experiment run 2. Experiment run 2 was performed one week after Experiment run 1. The initial stage of the experiment was the calibration of equipment such as the eye-tracker glasses and AidLab (ECG). After the calibration process, the main experiment began. The experimental procedure consisted of the four following steps (see Figure 3):

***Step 1.*** Biosensors (ET and ECG) calibration. In the case of eye-tracker glasses according to the instruction [20], only one point calibration was performed. For the wearable chest belt ECG, we were required to check that the belt was properly worn and that a strong signal was recorded. Calibration of biosensors was performed for both Experiment run 1 and Experiment run 2. ***Step 2.*** After the subject reported that he was ready to perform the experiment, the subject’s first task was to connect the patient to medical devices (saturation probe, blood pressure cuff, 3-lead). Step 2 was the same for Experiment run 1 and Experiment run 2.***Step 3.*** Monitoring the patient’s vital signs. In the step, we interfere in the vital sign level and the way they were displayed. Step 3 had different construction for Experiment run 1 and Experiment run 2.**Experiment run 1.** In this step, the medical staff was tasked with monitoring the patient’s vital signs. After 105 s, saturation went from 100% down continuously for 30 s.**Experiment run 2.** In this step, the medical staff was tasked with monitoring the patient’s vital signs. After 100 s, the random stressor (telephone ring or ambulance signal) was applied. The procedure aimed to induce stress in the subject. Five seconds after the stressor was turned on, the saturation went from 100% down continuously for 30 s. The stressor was turned off after 25 s from switching on, in order to calm the subject. During the falling vital signs of the patient and the triggering stimulus, the subject was obliged to undertake an appropriate medical procedure aimed at saving the patient’s life.***Step 4.*** When Experiment run 1 and Experiment run 2 were finished, the results from both experiments were analyzed for subject behavior, together with ET and ECG signals.

**Metrics for stress recognition.** The topic of stress detection has been reported in different studies. The most often analyzed signals for stress identification were eye-tracking and ECG signals. For ET signal, pupil diameter and mean fixation duration are the most frequently used metrics [21]. For the ECG signal, among used metrics, usage of the HR (ECG) and LF/HF (ECG) metrics seem promising [22]. According to [22], these metrics were reported as the most robust. Therefore, we used these metrics in our analysis to guarantee that the used stimulus for stress induction were measured during the experiment. Below we give the definitions of the metrics:**Mean fixation duration (ET)** measures how long a person’s attention was focused on the scene. The higher the stress, the shorter the mean fixation duration. The eye movement becomes more chaotic [23].**Pupil diameter (ET)** is a common characteristic used in stress detection [24]. If an individual’s pupil diameter increases [25,26], the pupil dilates at a higher frequency, then it suggests that the individual is possibly in a stressed state [21,27,28,29].**HR (ECG)** is a metric that measures heart rate variability, where the time between heartbeats varies slightly. Overall activity is more intense under stressful conditions [30,31,32,33].**LF/HF (ECG)**—Low frequency to high frequency ratio. It is an indicator of the autonomic nervous system balance between sympathetic to parasympathetic parts [34,35,36,37,38]. In the event of a stressful situation, the value of the metrics increases evoked by the reaction of the parasympathetic and sympathetic nervous system.

## 4. Results

The following section discusses results from the perceptual experiment to find the influence of the induced stress on the time of medical staff reaction during a patient’s life parameter monitoring. To assure that the stimulus used in the experiment induced a stress reaction, we computed the metrics from ET and ECG signals. To make sure that the stimulus used in the experiment caused the stress response, we plotted the raw data (notation: HR (ECG), LF/HF (ECG), Pupil diameter (ET), and Mean fixation duration (ET)) to assess the changes taking place on the charts visually. We then examined the statistical significance to check whether the responses indicated by the metrics differed statistically significantly between stressful and stress-free situations. The results are presented in Table 1, yielding *p*-value < 0.001 for all parameters. The analysis confirmed the consistency of the results and the stress response to the applied stressors. Change of pupil diameter, fixation time, heart rate, and LF/HF metrics are depicted in Figure 4. The averaged pupil size for all subjects indicates that the pupil was approximately 23% larger when respondent noticed a decrease in the patient’s vital signs for the situation with the switched-on stimulus than in the absence of the stress stimulus. Additionally, in a stressful situation, when the respondents noticed a decrease in vital parameters, a decrease in fixation time by an average of 47% was noticed in comparison to the stressless situation. The ECG record indicates an accelerated heart rate and increase of LF/HF metrics when the vital parameters decrease was noticed in the variant with the stressor in comparison to the stressless variant.

The next step was the analysis of the time of medical staff reactions to decreasing patient life parameters noticed in a stressless situation and under the induced stress. The difference is computed based on the eye-tracker data. To identify the moment when the respondent noticed the decreased saturation parameter, we defined the area of interest (AOI) when the parameter was displayed (see Figure 5). Visualization of the difference between the respondents’ gaze trajectories to reach the AOI in both stress and stressless situations is depicted in Figure 5 middle and Figure 5 right, respectively. In Figure 5 middle, the example of a respondent’s gaze trajectory in a stress-free situation is displayed. It is noticeable here that the subject quickly noticed a decrease in the patient’s vital parameters, and the trajectory of his gaze was not chaotic. The situation with the stressor switched on is shown in Figure 5 right. Here, the trajectory of the gaze is very chaotic, which is characteristic of a stressful situation. In this case, the subject took much longer to notice the decrease in the patient’s vital signs on the cardiac monitor.

The results of the one-way two-level Kruskal–Wallis test (with 5% significance level) for time reaction on decreasing patient life parameters in stressless and stress situations (with the stimulus that induced the stress) is equal to *p* = 0.001. The Kruskal–Wallis test analysis results significantly differ between analyzed reactions (see Figure 6).

## 5. Discussion

We have tested the hypothesis concerning the influence of stressors on the time of medical staff reaction on changing of a patient’s vital parameters. The hypothesis stated that stressors applied during simple clinical procedures concerning monitoring of vital signs will interrupt the correct actions of the research respondent. To receive reliable results, we ensured that the medical staff who took part in the experiment had similar medical backgrounds and experiences. We did not find any outliers in the subjects’ reactions. The results were entirely consistent, which made the results subject-independent. For the statistical analysis, we observed statistically significant prolonged response times of clinical decisions (*p*-value < 0.001). We also found clear signs that the participants’ stress-related behavior, i.e., increased pupil diameters, shortened fixation times, and heart rates, were consistent and significantly differed between situations with and without stressors (*p*-value < 0.001).

We noticed the confirmation of our hypothesis in publication [8]. However, although the authors emphasize that the effectiveness of an employee is affected by the lack of negative factors that make work difficult, their research did not consider the aspect of stress, which is a natural factor occurring in the emergency department, among other places. The authors also do not indicate that stressors cause a delay in actions taken by medical personnel aimed at saving the patient’s life.

The problem of stress in work as adverse events was reported by Waeschle et al. [9]. The authors argue that many adverse events could be avoided by excluding stressors. However, keep in mind that an emergency room is a place where stress is inevitable due to the characteristics of the job. Nevertheless, the authors of this publication did not refer to the reaction time in stressful situations compared to non-stress situations.

The results of mistakes analysis resulting from the usage of standard medical interfaces have already been subjected to discussion in different studies. The authors of several papers [1,4,11,12] have made it clear that many medical errors are made by inappropriate UI design. However, they did not refer to stressful situations. Discussing the issue of the impact of stress in the context of medical errors was done by MacDonald in [16]. The author pointed out that in stressful situations, there is a high probability of mental overload, which may lead to people to start behaving less rationally. However, in this publication, nothing is noted about the reaction time in situations where there is a stress factor.

According to our research, we conclude that stress during work in the emergency department delays the operation of medical personnel that used the standard set-up graphical interface. It means that the reaction time aimed at saving the patient’s life may be significantly longer. The result is of high importance and indicates that the standard user interface is ineffective in a real medical situation, quite different from the laboratory environment. That opens the area of research on the cardiac monitor interface, to make it efficient despite the medical staff’s cognitive load. The result is consistent with the Sittig et al. report [39] from 2020, where they indicated several key challenges to be addressed in the next 3–5 years. Developing standard user interface design features and functions to improve patient safety was one of them [39]. With medical interfaces standardized for clinical data input and output, further studies should address the adjustment of the interfaces (e.g., cardiac monitors) to the individual properties of the caregiver’s style of thinking.

In our study, we noticed that simple but frequently used medical procedures (monitoring basic patient vital signs) were already impaired by caregivers’ responses in a stressful environment. More complex medical procedures are likely to cause even more errors or delays in performance. In addition, the results of our research can be used as an introduction to further analyses concerning the identification of sensitive areas on the screen, where information is poorly perceived by users. Moreover, they can be confronted with cognitive properties of participants. This is our focus for future research.

## 6. Conclusions

The study results presented in the paper were the analysis of medical staff behavior during simple clinical procedure monitoring of vital signs that were interrupted with a stress-inducing stimulus, such as phone bell or the signal of an ambulance.

The significant difference between reaction time in stressless and stressful situations was measured during the user study that was performed with medical simulations. We observed prolonged response times of clinical decisions and clear signs of the participants’ stress-related behavior.

By studying both situations (stressless and stressful), we can formulate the statement that even the best-designed user interfaces, tested in a laboratory environment, i.e., without introducing the exhausting and annoying stimuli together, can be worthless. Human behavior, especially during stress and fatigue conditions, is determined by different factors. In our opinion, one of the most promising approaches can be the analysis of cognitive styles of subjects that describes natural human behavior, independent of his/her ability and knowledge.

## Figures and Tables

**Figure 1 sensors-22-03536-f001:**
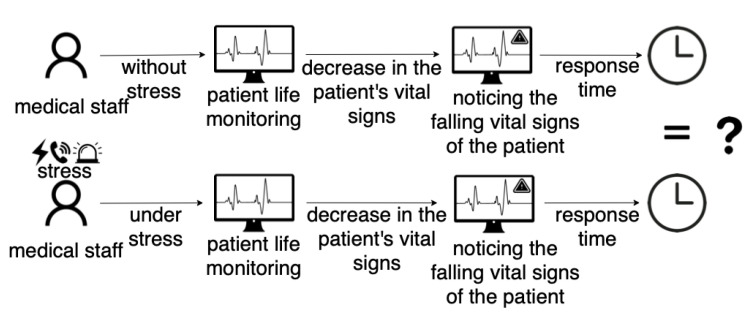
Problem definition. The question is about differences in the medical staff response time to the patient’s falling vital signs in a neutral (stress-free) and stressful situation.

**Figure 2 sensors-22-03536-f002:**
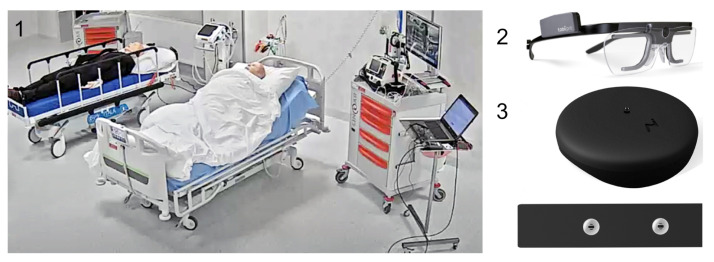
Experimental stand. **1**—Composed of a bed on which a phantom (patient) lies, a cardiac monitor (Terra Monitor), **2**—Tobii Eye Tracker Glasses 2 [18] with a laptop recording data, and **3**—the wearable chest belt for EEG signal recording (AidLab) [19]. The stand is located in the Medical Simulation Center of Collegium Medicum of the University of Zielona Gora.

**Figure 3 sensors-22-03536-f003:**
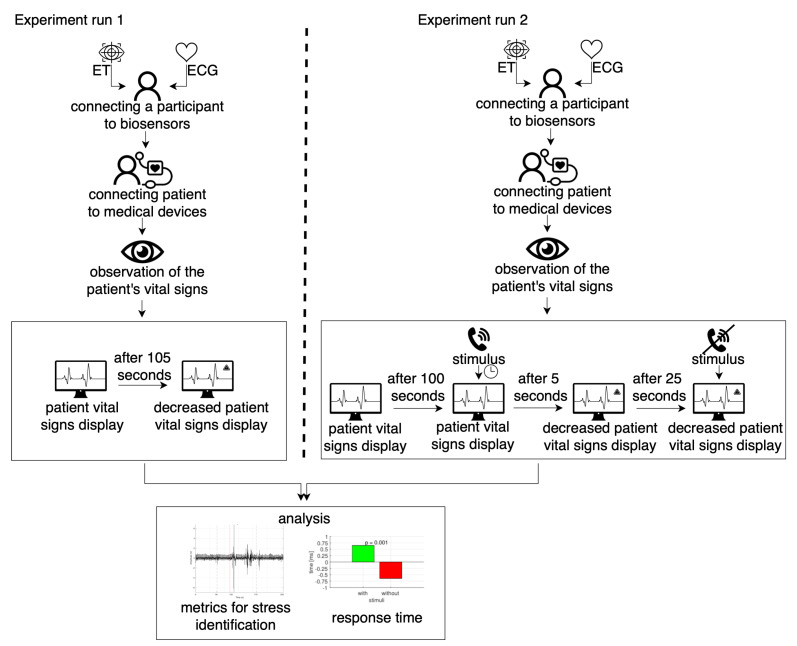
Experiment overview.

**Figure 4 sensors-22-03536-f004:**
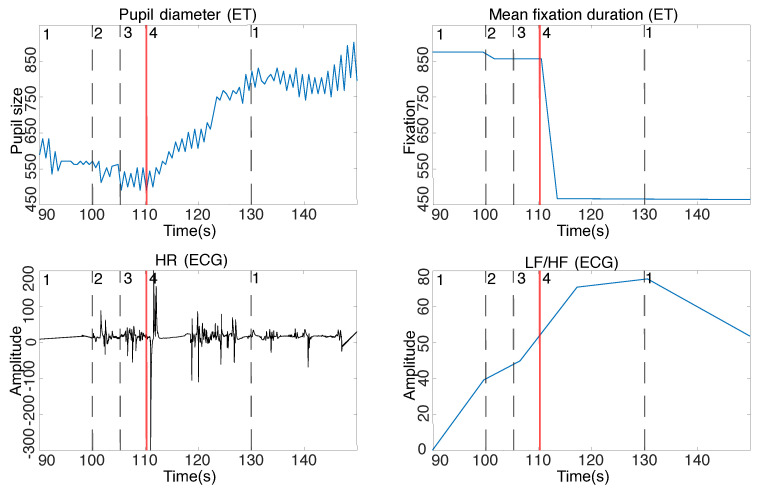
**Top Left**: change of pupil diameter; **Top Right**: change of fixation time; **Bottom Left**: ECC signal—change in heart rate due to ECG recording; **Bottom Right**: change of LF/HF metrics from ECG recording (sympathetic to parasympathetic response). **1**—the moment of the experiment start (without the use of the stressor), **2**—the moment when the patient’s vital signs began to drop on the monitor of the patient’s life, **3**—the stressor swiched on, **4**—the moment when the subject noticed the falling vital signs of the patient on the cardio monitor.

**Figure 5 sensors-22-03536-f005:**
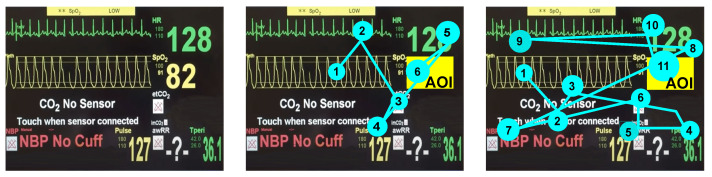
**Left**: Cardiac monitor used to perform the experiment. **Middle**: The course of the fixation trajectory of the respondents in a stress-free situation. **Right**: The course of the fixation trajectory of the respondents in a stressful situation. AOI—Area of interest (target place on the cardio monitor, where the patient’s vital signs fell), cyan spots—show the course of the subjects’ eye trajectories in the order from [1 to n].

**Figure 6 sensors-22-03536-f006:**
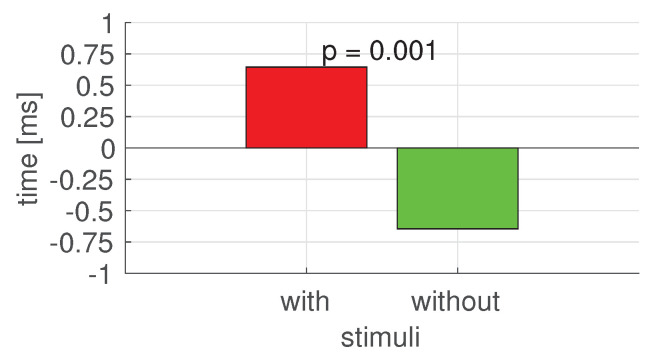
The difference in mean time reaction between situation with stressful stimulus (red) and without stressful stimulus (green). The *p*-value indicates the statistical difference between compared stimuli, computed by the Kruskal–Wallis test at a 0.05 significance level.

**Table 1 sensors-22-03536-t001:** List of the metrics on the basis of which the stress was detected with the Kruskal–Wallis test value.

Metrics	Kruskal–Wallis Test
Mean fixation duration (ET)	*p*-value < 0.001
Pupil diameter (ET)	*p*-value < 0.000
HR (ECG)	*p*-value < 0.000
LF/HF (ECG)	*p*-value < 0.001

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
