# Peer review of "Can the Standard Configuration of a Cardiac Monitor Lead to Medical Errors under a Stress Induction?"

_sensors, 2022, doi:10.3390/s22093536_

Round 1
Reviewer 1 Report
The presented work describes a study that analyzes the behavior of medical staff in inducing a stressful situation. However, the value and significance of this study are somehow reduced by the fact that the induction of a stressful situation takes place in artificial conditions. The participants in the experiment are aware that if they make a mistake, nothing significant will happen. However, such a situation may be distant from the actual clinical operation. The result of the finding is basically that in the induced stress there is a delay in the reaction time of the operator. This is a pretty familiar thing to me. A novelty that can lead to valuable results is the use of a vision tracking system, but only to simply capture the trajectory of observation. At the same time, the study could bring much more if the authors proposed improvements in the layout of the display area in order to improve clarity and eliminate the tedious chaotic search for key information.
The Problem definition in Figure 1 may not be obvious without an explanation in the text. The figure is rather confusing.
Minor spell check is required: „patent’s monitor” , “people?s“, „the Kruskal–Wallis rest“
The list of literature is quite extensive, however, I recommend the authors to study also e.g.
Cvach M.: Monitor alarm fatigue: an integrative review. Biomed Instrum Technol. 2012 Jul-Aug;46(4):268-77. doi: 10.2345/0899-8205-46.4.268
Reviewer 2 Report
This paper studied the medical staffs' reaction in a stressful situation. The heart rate (ECG signals) and eye-tracker (ET signals) were used as the biomedical censoring inputs. The parameters of mean fixation duration, pupil diameter, heart rate, and low frequency to high frequency ratio were utilized to analyze the stress induction of the patient's monitor. The following comments may be considered for further improvements:
1) The title of Section 2: "Future works" should be revised. The introductory paragraphs typically appears in the research background presentation section. Future works are mentioned in the conclusive section.
2) In the bottom left column of Figure 4: it seems not an ECG signal. The figure caption states the change of heart rate, but the figure title shows as "ECG signal". Please clarify what exactly the signal is.
3) How to detect the changes of the ECG or ET signals related to the event of stressor switch on? Please provide the algorithms or procedures used for stress event detection based on ECG or ET signals.
4) In addition to the mean time reaction difference as shown in Figure 6, it is necessary to display the differences of the other parameters in the figure. The statistics of the parameters in stress situation versus normal situation should be tabulated, with the p-values of the Kruskal-Wallis tests.
5) The discussion section may include the discussions on whether the statistical difference results were subject-dependent or subject-independent.
